# Annotation-free delineation of prokaryotic homology groups

**Yongze Yin**[1]*, **Huw A. Ogilvie**[1]*, **Luay Nakhleh**[1,2]*

**1** Department of Computer Science, Rice University, Houston, Texas, United States of America,
**2** Department of BioSciences, Rice University, Houston, Texas, United States of America

* yongze@rice.edu (YY); huw.a.ogilvie@rice.edu (HAO); nakhleh@rice.edu (LN)

## Abstract

Phylogenomic studies of prokaryotic taxa often assume conserved marker genes are homologous across their length. However, processes such as horizontal gene transfer or gene duplication and loss may disrupt this homology by recombining only parts of genes, causing gene fission or fusion. We show using simulation that it is necessary to delineate homology groups in a set of bacterial genomes without relying on gene annotations to define the boundaries of homologous regions. To solve this problem, we have developed a graph-based algorithm to partition a set of bacterial genomes into **Maximal Homologous Groups** of sequences (**MHGs**) where each MHG is a maximal set of maximum-length sequences which are homologous across the entire sequence alignment. We applied our algorithm to a dataset of 19 Enterobacteriaceae species and found that MHGs cover much greater proportions of genomes than markers and, relatedly, are less biased in terms of the functions of the genes they cover. We zoomed in on the correlation between each individual marker and their overlapping MHGs, and show that few phylogenetic splits supported by the markers are supported by the MHGs while many marker-supported splits are contradicted by the MHGs. A comparison of the species tree inferred from marker genes with the species tree inferred from MHGs suggests that the increased bias and lack of genome coverage by markers causes incorrect inferences as to the overall relationship between bacterial taxa.

## Author summary

Assuming genes to be the basic evolutionary unit has been commonplace in bacterial genomics. For example, when quantifying the extent of horizontal gene transfer it is common to infer gene trees and reconcile them against a species tree to account for recombination-based processes. We have developed a new method which challenges this assumption by identifying contiguous regions of true homology without regards to gene boundaries and applied it to Enterobacteriaceae, a family of bacteria containing several important human pathogens. Our results show that genes are composed of distinct homologous regions with conflicting phylogenetic histories. We further demonstrate that failing to take account of this conflict, together with the functional biases we show exist

**Data Availability Statement:** The code for our implementation is available at https://github.com/NakhlehLab/Maximal-Homologous-Groups along with instructions. While we did not generate any original sequence data for this manuscript, the NCBI accession numbers for the genomes used in

this manuscript are GCF_000973545.1, GCF_900039485.1, GCF_900048035.1, GCF_900044015.1, GCF_000828515.1, GCF_000093065.1, GCF_000828815.1, GCF_000757825.1, GCF_000648515.1, GCF_000982825.1, GCF_000164865.1, GCF_000299455.1, GCF_001887595.1, GCF_001022135.1, GCF_000300455.3, GCF_000757785.1, GCF_000195995.1, GCF_000006925.2 and GCF_000262305.1.

**Funding:** This work was funded in part by National Science Foundation (https://nsf.gov/) grants DBI 2030604, CCF 1514177, CCF 1800723 and EF 2126387(to L.N.). The funders had no role in study design, data collection and analysis, decision to publish, or preparation of the manuscript.

**Competing interests:** The authors have declared that no competing interests exist.

among single-copy marker genes, significantly changes the consensus evolutionary tree of Enterobacteriaceae.

This is a *PLOS Computational Biology* Methods paper.

## Introduction

Studying prokaryote evolution through the lens of comparative genomics has been conducted using one set of approaches at deep timescales and another set at shallow timescales. At deep timescales where genes and genomes are highly diverged, copies of marker genes which tend to be conserved across many species are identified in the genomes being studied. A marker gene is defined as a protein-coding gene which ideally occurs as a single-copy gene in every studied genome. Multiple sequence alignments (MSAs) of each marker gene may be analyzed individually [1], or concatenated into a supermatrix and be analyzed together [2].

While early phylogenetic studies relied on one or two marker genes such as SSU rRNA [3], the advance of sequencing technologies including next generation sequencing and more recently single molecule real time sequencing has enabled the transition to phylogenomics and the construction of datasets with hundreds of genes from entire genome assemblies [1, 2, 4]. Despite the availability of whole genomes, it remains the default assumption to work with functional genes as the basic evolutionary unit for studying evolution [5]; in a typical phylogenomic analysis where a species evolutionary history is inferred from the genealogies of individual loci, a central mathematical assumption is that the evolutionary history of each locus is recombination free.

However, when functional genes are used as loci, this ignores the possibility that recombination may break up the phylogenetic history of different segments within a gene. This problem is well known with regards to eukaryotes, where meiotic recombination breaks up these histories with each generation. In that case, functional genes are known as "m-genes" and maximal segments which share a common phylogenetic history are known as coalescence genes or "c-genes" [6, 7]. These c-genes are defined as "a segment of the genome for which there has been no recombination over the phylogenetic history of a clade" [6], and it has been previously shown that the species phylogeny can be incorrectly inferred when concatenating multiple c-genes [8]. Identifying c-genes can be very challenging [9], which has led to the common phylogenomic practice of using loci that are individually short and are far apart, even if this practice does not guarantee that the loci constitute c-genes. When the genomes or genomic regions under analysis are collinear and alignable, methods such as coalescent hidden Markov models can be used to delineate c-genes [10], though these methods suffer from scalability issues [11].

While prokaryotes do not undergo meiotic recombination, they experience many other forms of recombination such as horizontal gene transfer and gene duplication/loss. These events are often non-homologous, disrupting even looser assumption of homology along the entire length of functionally equivalent genes. It has been shown that genomic regions with different lengths can be impacted, from a single nucleotide [12], multiple nucleotides [13, 14], a gene [15, 16], an operon [17–20], to even an entire chromosome [21]. Therefore, it is important both not to treat a gene as an evolutionary unit, and to identify homology groups without

relying on marker gene annotations (homology groups were referred to as "module families" in [22]).

To understand the potential for erroneous phylogenomic inferences when "m-genes" are used instead of "c-genes" for inferring prokaryotic phylogenies, we conducted a simulation study where markers were simulated from multiple c-genes differing slightly along their length. Our results will show that this leads to extremely high bootstrap support for incorrect clades. Work on this problem has been done, but is based on proteomes derived from genome annotations rather than whole genomes. Both the methods of Wu *et al.* [22] and Leonard and Richards [23] are based on processing BLASTp queries to identify MHG boundaries within protein sequences, and will truncate MHGs that extend past coding regions, and may be inaccurate where errors exist in genome annotations, or where mutations are present that disrupt the coding sequences such as frameshifts or changes to stop codons.

Whole genome aligners may be used at shallow timescales to align as many homologous pairs of nucleotides as possible across input genomes. This approach reveals the relationship between different strains within a species or species complex, and the forces of molecular evolution shaping their genomes [24, 25]. While much more of the input genomes are used in this approach compared with marker genes, due to the high degree of divergence between bacteria species, state-of-the-art whole genome aligners all have difficulties at deeper timescales. For example, Parsnp [26] aligns only core genomes defined as orthologous sequences conserved by all input genomes. ProgressiveMauve [27] is extended from mauve [28] utilizing k-mer based locally collinear blocks, which is significantly slower and consumes more space when faced with highly divergent genomes. Progressive Cactus [29] requires a guide tree and performs well when aligning different samples of the same species, however its performance is questionable when aligning bacterial genomes of different species, which will be much more divergent than genomes of different primates or other mammals. SibeliaZ [30] is a de Bruijn based whole genome aligner that is faster than Progressive Cactus but has an even smaller tolerance of divergent input genomes.

Here we introduce a method that identifies groups of sequences that share common ancestry along their entire length. It is based on constructing an graph of intervals connecting segments of different genomes which have evidence for local homology at the nucleotide level, initializing each segment as a "MHG," and walking the graph to merge and partition MHGs. The end result is a set of MHGs where each MHG contains a set of homologous sequence segments. Sequences assigned to each MHG using our method may be orthologous or paralogous, in the latter case due to hidden paralogy and because we incorporate every sequence identified as homologous from each genome, which necessarily includes gene duplications. Gene losses may result in zero sequences being included from a given genome.

Compared with the marker gene approach for deep timescales, our method has the advantage of using more of the input genomes when the divergence is low enough for sequence homology to be identifiable at a nucleotide level. The selection of marker genes is biased towards slow evolving genes, since genes with frequent point mutations will be difficult to align, and genes with structural variations should not be aligned as a single unit. These slow evolving markers may not be representative of the whole genome in terms of their structure and function, and we report evidence for this bias in our study and show how our MHGs compiled by our method by contrast are more broadly representative.

Another advantage over marker gene studies is that our algorithm does not rely on any existing genome annotation to define the boundaries of the sequences (e.g., by referring to previously inferred boundaries of domains, coding sequences or operons). This compares with a previously published method for compiling MHGs which takes protein sequences as input instead of a genome assembly [22]. That method was also targeted at *Drosophila*, a genus of

eukaryotes, where evolution is much slower and hence the difficulty of the problem is easier than for prokaryotes.

To analyze the performance of our method, we applied it to a select subset of Enterobacteriaceae genomes. We found that our algorithm is able to cover most of the genes with MHGs for free-living bacteria other than the presumptive outgroup taxon, especially compared with the limited number of marker genes. A further analysis of gene ontology (GO) terms associated with marker and MHG-covered genes showed that MHGs are less biased in their associated functions than marker genes are. Finally, a phylogenetic analysis revealed concerning levels of conflict between the marker genes and associated MHGs, suggesting that marker genes are an inappropriate unit for phylogenomic studies and may infer false evolutionary relationships. Our new method for the preparation of more reliable MHG-based collation of phylogenomic data is available at https://github.com/NakhlehLab/Maximal-Homologous-Groups.

## Materials and methods

### Algorithm for delineating maximal homologous groups of sequences

Given a set of genomes, we define a maximal homologous group of sequences (MHG) as a maximal set of maximum-length sequences (substrings of the genomes) whose evolutionary history is a single tree and involves no rearrangements. The set is maximal in that adding more sequences to it violates the single-tree assumption and the sequences are of maximum lengths in that including more nucleotides in them also violates the single-tree assumption (e.g., having fused genomic regions). MHGs are defined for any set of two or more genomes, with no regard to functional annotations.

A high-level description of the algorithm is described below and illustrated in Fig 1. A full implementation of the algorithm is available at https://github.com/NakhlehLab/Maximal-Homologous-Groups.

Our algorithm uses graphs in two key ways. Firstly, the alignment graph is used to store the connections between genome segments identified using BLASTn in an efficient way, and since segments from different connected components of the graph will never be merged into the same MHGs, the partition and merge step may be parallelized by connected component, or flushed to storage after each connected component is traversed to reduce memory usage. Secondly, MHGs are stored as graphs in order to propagate boundaries between genomes while taking into account indels, as described below. Starting from a set of genomes:

1. **All-versus-all BLASTn to find pairwise local alignments** (Fig 1B) All versus all BLASTn queries are filtered based on the bit score values [1]. For each genome, the maximum possible bit score is calculated from the set of results obtained by querying the genome against itself using BLASTn. All versus all BLASTn results are subsequently excluded if their bit score falls below a certain threshold of the maximum bit score obtained for the genome the query sequence derived from.

2. **Sequence pile-up** (Fig 1C) Given every pairwise alignment call by BLASTn, the algorithm piles up alignments that cover overlapping regions on the same genome.

3. **Initial MHGs and alignment graph construction** (Fig 1D and 1E) For piled-up alignments in every genome, the algorithm takes the union of overlapping alignments resulting in a set of non-overlapping ranges in all genomes. An alignment graph is constructed from adding the range set as nodes, and an edge is added for each local alignment to connect the two nodes that the subject and query ranges of the local alignment correspond to

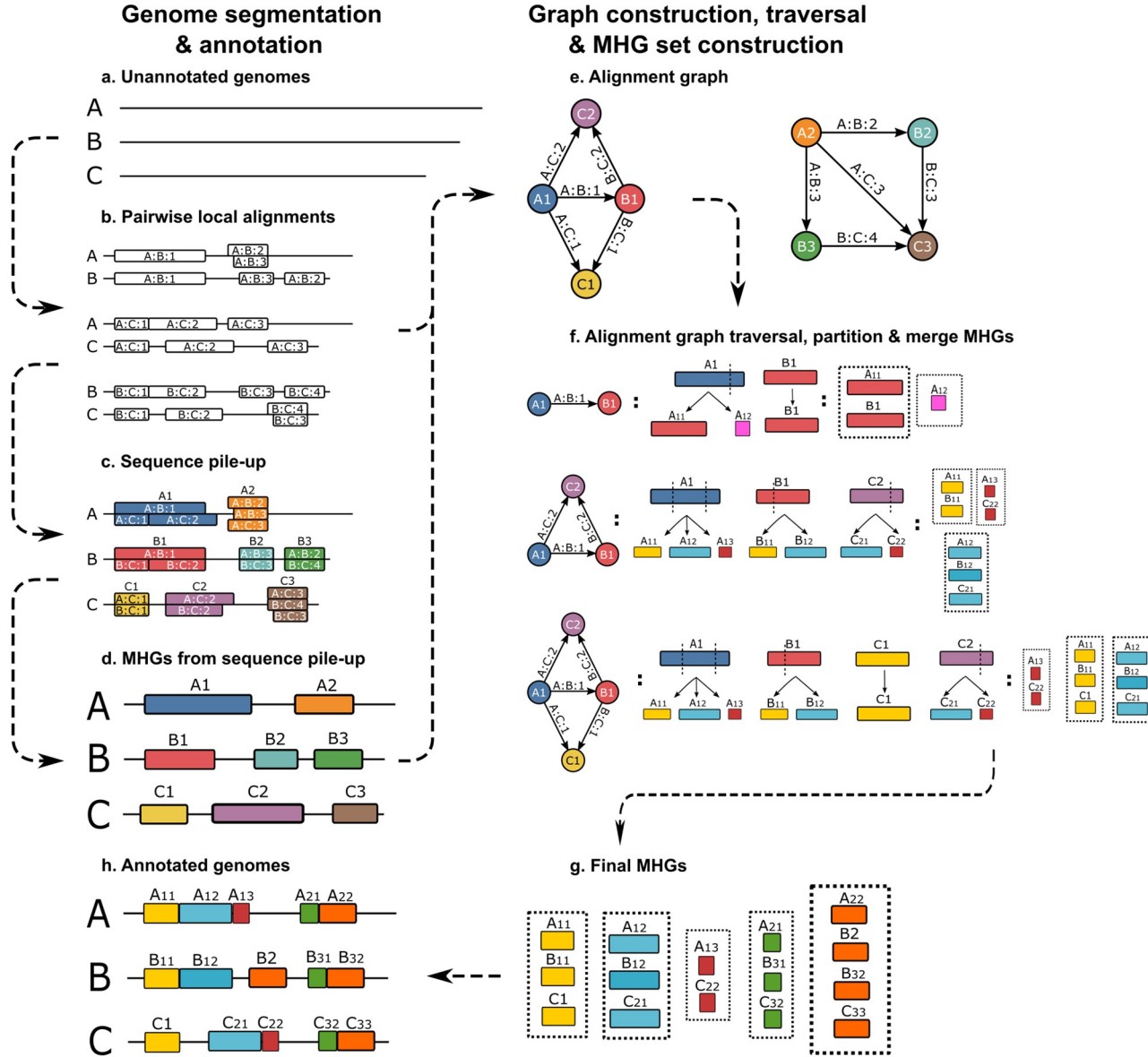

**Fig 1. Algorithm for delineating maximal homology groups of sequences (MHGs).** (a): The input consists of a set of two more genomes. (b): Pairwise local alignments are computed. (c): The input to our algorithm is a set of pairwise local alignments (b) found in whole genomes of related taxa (a). Our algorithm piles up the segments within each genome corresponding to the pairwise local alignments (c) to construct a set of initial MHGs, each with a single sequence (d). An alignment graph (e) is constructed where the edges correspond to pairwise local alignments, connecting nodes corresponding to initial MHGs. As each connected component of the graph is traversed, MHGs are merged based on being connected by pairwise alignments, and partitioned based on the boundaries of those alignments, and the boundaries of the MHG they are being merged with. Traversal of one of the connected components (f) is illustrated by the edges being traversed (left), the resulting partitioning (center) and merged MHGs (right). Once the traversal is complete, the final set of MHGs (g) can be used to annotate the genome sequences with homology groups (h). Dashed arrows show the data input and output relationship between stages.

(There could be multiple edges between two nodes). The alignment graph serves to efficiently split up a big problem to a few subproblems where each connected component in the alignment graph can be traversed independently. In other words, A MHG can only include subsequences of nodes from the same connected component. Encountering a

new pairwise alignment mapping, the algorithm only needs to check and partition MHGs generated from the same connected component instead of searching around full set of MHGs.

4. **Alignment graph traversal** (Fig 1F) Starting from the the alignment graph, every edge is visited using a depth-first search (DFS) of every connected component. Whenever a new breakpoint is inserted, MHGs containing segments which include that breakpoint will be split into two at the breakpoint, and pairs of post-split MHGs connected by the local alignment edge will be merged. For example, when visiting edge A:B:1 in Fig 1F, encountering a first breakpoint carried by $B1$, MHG $A1$ is partitioned into $A_{11}$, $A_{12}$, and $A_{11}$ is merged with $B1$ because edge A:B:1 connects $A_{11}$ and $B1$. The split and merged MHGs will contain directions for all sequences based on the relative directions of the subject and query sequences in the local alignment (i.e. in the same direction on both genome assemblies, or opposite directions, relative to the strand represented by those assemblies). And after visiting all edges in the alignment graph, the output MHGs will contain the homology groups for the input whole genomes.

When a new breakpoint is added to an existing MHG after processing a local alignment, we use the pairwise local alignment estimated by BLASTn to ensure the new breakpoint is inserted at accurate locations. Our algorithm converts the pairwise local alignments to two bit arrays where 1 means a match, and 0 means a gap. We first locate the correct location of the breakpoint in a pairwise alignment by counting 1s from the beginning of the bit array for the source genome (for which already know the breakpoint location), to match the difference between the breakpoint coordinate in the source genome and the starting coordinate of that local alignment. Second, for the destination genome (where the breakpoint location is unknown), we count 1s in its bit array, going backwards from the breakpoint location (which is the same as identified in the source bit array) until we reach the beginning of the bit array. This then gives us the difference in position between the starting coordinate of the local alignment in the destination genome and the breakpoint.

The first stage, all-vs-all BLAST search has a time complexity $O(n^2)$ where n is the number of genomes, since every genome is queried by every genome. In a hypothetical scenario where all genes are present as single-copy orthologs in every genome, the alignment graph to be walked in the second stage will have $O(n^2)$ edges. Since every edge is traversed once, this stage also has a time complexity of $O(n^2)$. When an edge is visited that leads to an ortholog from a new species being added to an existing MHG, this genome will be segmented at that locus and the ortholog incorporated into the MHG through backtracking. A new R-tree [31], used to store the segmentation of each genome, will need to be constructed which in the worst case takes $O(s \cdot log(s))$ time where $s$ is the number of segments, which will be proportional to the number of single-copy orthologs. The backtracking process is of time complexity $O(m)$ where $m$ is the number of sequences in the MHG, which will grow with $n$. However, as $n$ increases, an increasing proportion of edges will not lead to the incorporation of new sequences, hence R-tree construction and backtracking can be discounted.

For any edge visited, the R-trees of the source and destination genomes are queried. The time complexity of these queries is bounded by $O(log(s))$ and $O(s)$, but since this will not depend on the number of genomes in this hypothetical scenario, the overall time complexity relative to the number of genomes is $O(n^2)$. Under more realistic conditions, the time complexity becomes much more difficult to analyze as the number of segments in each genome will depend on both the number of genomes and multifarious evolutionary processes.

## A prokaryotic dataset

We applied our method to a set of 19 bacterial species, each from a different genus of $\gamma$-proteobacteria; 18 from within Enterobacteriaceae and one outgroup. We chose the specific genome assemblies from a previously compiled dataset of 10,575 bacterial and archaeal genomes [2]. The assembly we used for the outgroup taxon has the RefSeq identifier GCF_001887595.1, which in the analysis of Zhu *et al.* [2] was mislabelled as *Klebsiella michiganensis* when it actually is *Luteibacter rhizovicinus* [32]. The genome assembly GCF_001022135.1 that we used for *Phytobacter ursingii* was originally identified as *Kluyvera intermedia* when it was first published but has since been reclassified [33, 34]. The species names and NCBI reference IDs used in our analysis are: *Blochmannia* endosymbiont (GCF_000973545.1), *Candidatus* Doolittlea endobia (GCF_900039485.1), *Candidatus* Gullanella endobia (GCF_900048035.1), *Candidatus* Hoaglandella endobia (GCF_900044015.1), *Candidatus* Ishikawaella capsulata (GCF_000828515.1), *Candidatus* Riesia pediculicola (GCF_000093065.1), *Candidatus* Tachikawaea gelatinosa (GCF_000828815.1), *Cedecea neteri* (GCF_000757825.1), *Citrobacter freundii* (GCF_000648515.1), *Cronobacter sakazakii* (GCF_000982825.1, *Enterobacter lignolyticus* SCF1 (GCF_000164865.1), *Escherichia coli* (GCF_000299455.1), *Luteibacter rhizovicinus* (GCF_001887595.1), *Phytobacter ursingii* (GCF_001022135.1), *Kosakonia sacchari* (GCF_000300455.3), *Pluralibacter gergoviae* (GCF_000757785.1), *Salmonella enterica* (GCF_000195995.1), *Shigella flexneri* (GCF_000006925.2), and *Shimwellia blattae* (GCF_000262305.1).

A marker gene is defined in an ideal sense as a single copy gene which occurs in every studied genome. In practice this requirement is too strict to construct a useful data-set with, and is relaxed from "every" to "most" genomes. In this paper, marker genes were previously identified protein-coding genes [2], and we used the best tBLASTn [35] hit from the proteome translated from the corresponding genome to identify the marker gene for that species, but only where it was identified as present in that species in the original study. Reference IDs for each marker gene from the original publication were used to retrieve the query protein sequences from UniProt [36]. And in order to calculate whether a marker gene is covered by any MHG, we used the best hit from tBLASTn to locate the nucleotide sequence in each genome corresponding to every known marker gene protein sequence. Marker coverage of each genome and its genes was calculated based on the full lengths of the specific genes identified in that genome, as above.

We constructed a nucleotide BLAST database from all 19 genomes to conduct the all verses all BLASTn search. For the search, we used a word size of 9, a gap open penalty of -1, and a gap extend penalty of -1. These parameter values were chosen so that our method will work even for very divergent genomes. A shorter word size than the default enables BLASTn to align regions sharing shorter exactly matched k-mers, and choosing the smallest possible mismatch and gap penalties should maximize the lengths of locally homologous alignments identified by BLASTn.

Next, we used the algorithm described above to infer the set of MHGs for those 19 genomes. Coverage of genomes and genes by MHGs was computed using BEDtools [31]. Since MHGs are non-overlapping, base pair coverage refers to the proportion of a genomic sequences covered by any MHG.

## Simulation of marker genes

To simulate marker genes with internal recombination we began with the species phylogeny previously inferred by concatenating marker genes [2] which was pruned to contain the above 19 bacteria. For each of 10 replicate marker genes, the first component tree was obtained by

performing a single nearest-neighbor interchange (NNI) operation on the initial tree, the second component tree was obtained by performing a single NNI operation on the first component tree, and so on. We chose NNI as a reasonable and conservative facsimile of horizontal recombination, since it will only exchange segments between closely related taxa.

We simulated 15 such component trees, generated 100bp sequence alignments with equal transition rates and base frequencies using Seq-Gen [37] on each component tree, and concatenated them to produce the marker MSA for a total length of 1,500bp, chosen because it is close to the average length of marker genes included in our study, which we calculated as being 1,503bp. The choice of 100bp was made to ensure phylogenetic histories of each segment were to an extent still resolvable while simulating a substantial number of recombination events within each gene, and equal rates and frequencies used since this study does not focus on the effects of varying substitution models or parameter values.

IQ-TREE [38] was used to infer maximum likelihood trees with bootstrap support for the 100bp non-recombining segments and the simulated markers with internal recombination. We pre-specified the HKY substitution model for IQ-TREE, of which the true model is a special case. Robinson–Foulds (RF) distances between known and inferred trees were calculated using phangorn [39].

## Phylogenetic analysis

For each marker and for each MHG containing at least 4 segments, we ran the FFT-NS-i [40] algorithm implemented in MAFFT with 500 maximum iterations. For each resulting MSA, IQ-TREE [38] was run with default settings to build a gene tree. Finally we used ASTRAL-Pro [41] to infer a species tree from the marker gene trees, and another species tree from the MHG gene trees, because this method accounts for gene duplication and our MHGs may contain more than one sequence per genome, and it has been shown to be robust to duplication and loss rates [42].

Because methods for quantifying phylogeny conflict are based on comparing trees with uniquely labeled tips, we excluded MHGs with more than one sequence derived from any single genome. For the nucleotide sequences of marker genes and overlapping MHGs with at least four taxa, an original tree and 100 bootstrap trees are inferred using the IQ-TREE maximum-likelihood method [38]. MHGs with an internal branch length less than 0.0005 in their original trees were regarded as unresolved and filtered out. This threshold value is smaller than the reciprocal of the average marker gene length of 1,503bp, so branch lengths below this threshold imply no mutations occurred along this branch (at least under a Jukes–Cantor model) and the relationship between the four subtrees adjacent to this branch is impossible to resolve. After removing ambiguities, every bootstrap tree in every marker and MHG will yield a set of bipartition splits for each internal node. We used DendroPy [43] to calculate bipartitions, which DendroPy encodes as bitmasks. Since a marker and its overlapping MHGs may consist a different set of taxa, a bitwise normalization operation is performed to ensure each bipartition is represented by the same bitmask, and by exactly one bitmask, for the marker and all overlapping MHGs.

## Functional annotation

To classify genes—either marker genes, or a gene associated with a MHG because they are overlapping—by function, we used a GO slim. This aggregates gene classifications into a small enough set of categories to be interpretable and reveal any bias in the distribution of gene function. We used a custom script to traverse the GO hierarchy, and categorized genes based on whether any of their GO terms had an "is_a" or "part_of" relationship to any one of the

Alliance for Genome Resources (ARG) GO slim terms [44]. We allowed multiple categories per gene when their GO terms mapped to multiple ARG slim terms.

## Results

### Simulation study

To understand whether it is better to use marker genes or MHGs as the basic unit of evolutionary inference, we inferred maximum likelihood trees from simulated marker genes, their non-recombining segments, along with bootstrap trees for both. The typical bootstrap support for branches inferred from simulated markers was much higher than component trees (Fig 2A and 2D) due to the longer length and higher information content. However this is deeply misleading, as the normalized Robinson-Foulds (RF) distance between the inferred marker gene tree and the true component trees is centered around 0.5, meaning about half the branches are incorrect (Fig 2C). This is worse than the accuracy of the trees inferred individually for each non-recombining component, which was typically below 0.5 (Fig 2F). For the non-recombining segments, bootstrap support for true branches was higher than incorrect branches, suggesting that bootstrap support values for inferred MHG trees can be used to identify true splits in the local phylogeny, as opposed to the clonal frame (Fig 2B and 2E).

### MHG occurrence across genomes

The dataset we analyzed consists of both free-living and endosymbiotic bacteria. Severe gene loss is common among endosymbionts [45], and as a result relatively few MHGs contain sequences from those species (Fig 3). In addition to the explicitly labeled Blochmannia endosymbiont, these species include all *Candidati*.

In terms of the coverage of genomes by MHGs at a nucleotide level, a substantial fraction of genes were covered by MHGs, even when we limited the MHGs to those with sequences from at least eight species (Fig 4). The difference between gene length and genome size was roughly

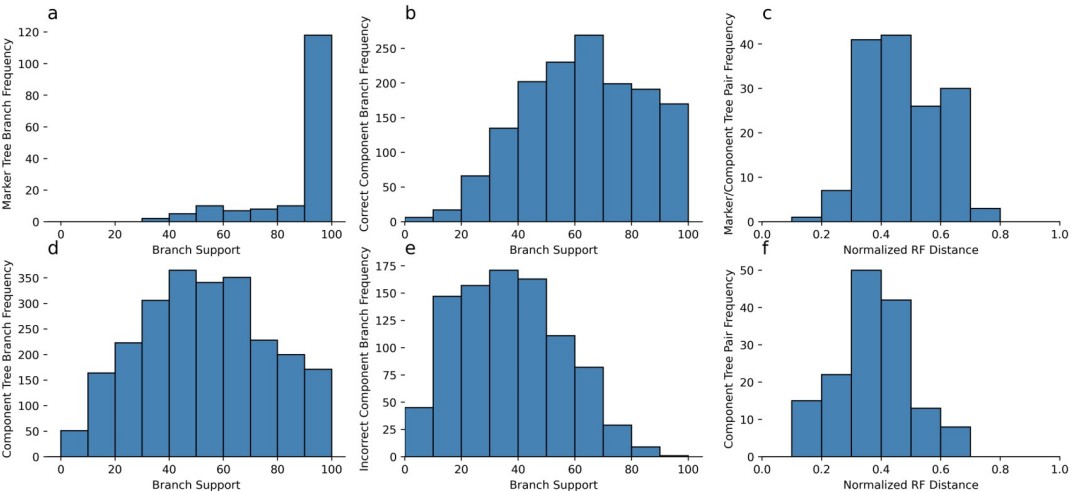

**Fig 2. Support for branches in simulated marker genes and component trees.** (a): Bootstrap supports for internal branches of inferred marker trees. (b): Bootstrap supports for correctly inferred component tree branches. (c): Normalized RF distances for every component tree and the corresponding inferred marker tree. (d): Bootstrap supports for internal branches of inferred component trees. (e): Bootstrap supports for incorrectly inferred component tree branches. (f): Accuracy of inferred component trees computed as normalized RF distances from the true component trees.

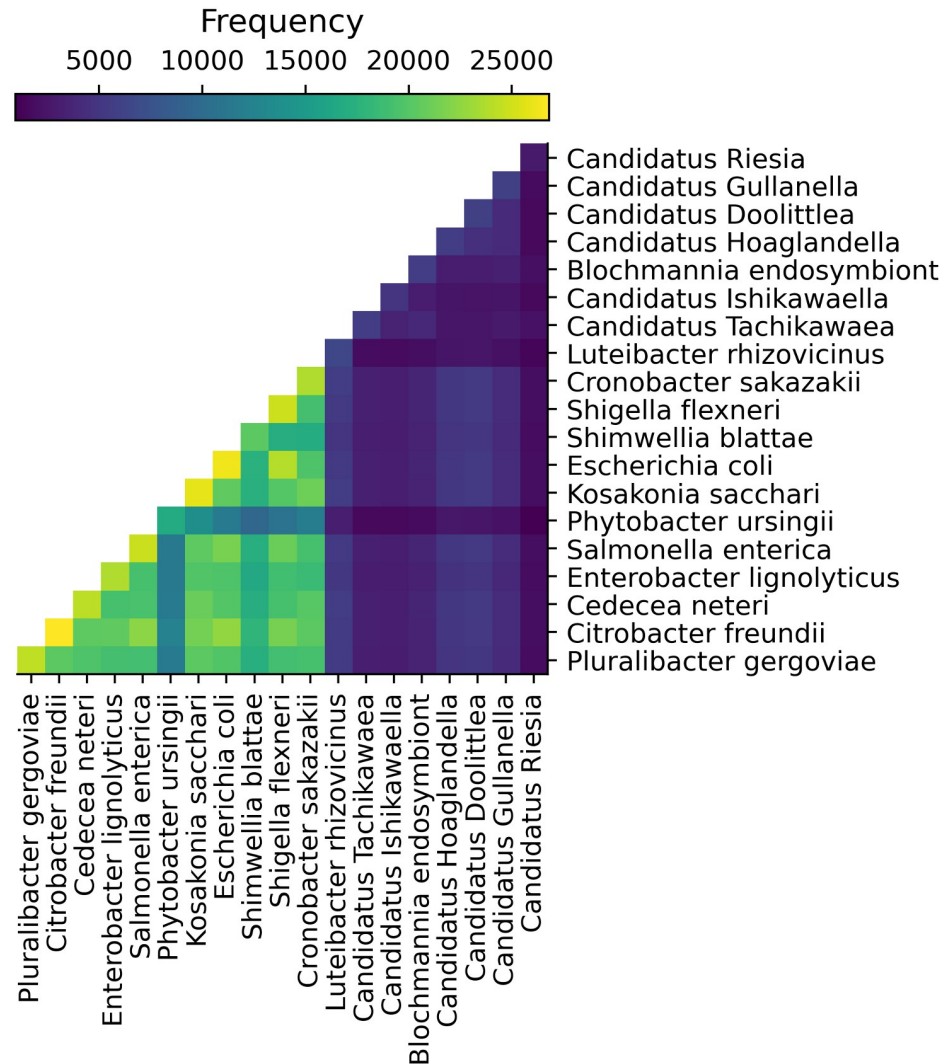

**Fig 3. Occurrence of MHGs by species.** Diagonal cells show the number of MHGs containing at least one sequence from the genome assembly for that species. Off-diagonal cells show the number of MHGs containing at least one sequence from the genome assemblies of both species, i.e. the pairwise occurrence.

proportional across genomes, and the gap between the two corresponds to the amount of intergenic sequence in each genome. Unlike eukaryotes, prokaryotes have diminutive genomes with consistently high gene density [46, 47].

Intergenic coverage was also good for free living bacteria other than *Luteibacter rhizovicinus*, but not for endosymbionts, indicating that endosymbionts have undergone substantial disruption to their genomes between as well as within genes. Because of the gene loss among endosymbionts, many genes in free living bacteria bear no homology to genes in endosymbionts, and therefore relatively few genes in free living bacteria are covered by MHGs with sequences from 16 or more species (Fig 4).

Fewer MHGs contained sequences from *L. rhizovicinus*, although the proportion of genic sequences covered by MHGs converged with other taxa for MHGs with more genomes included (Figs 3 and 4). By contrast, there was almost no intergenic coverage, even for MHGs

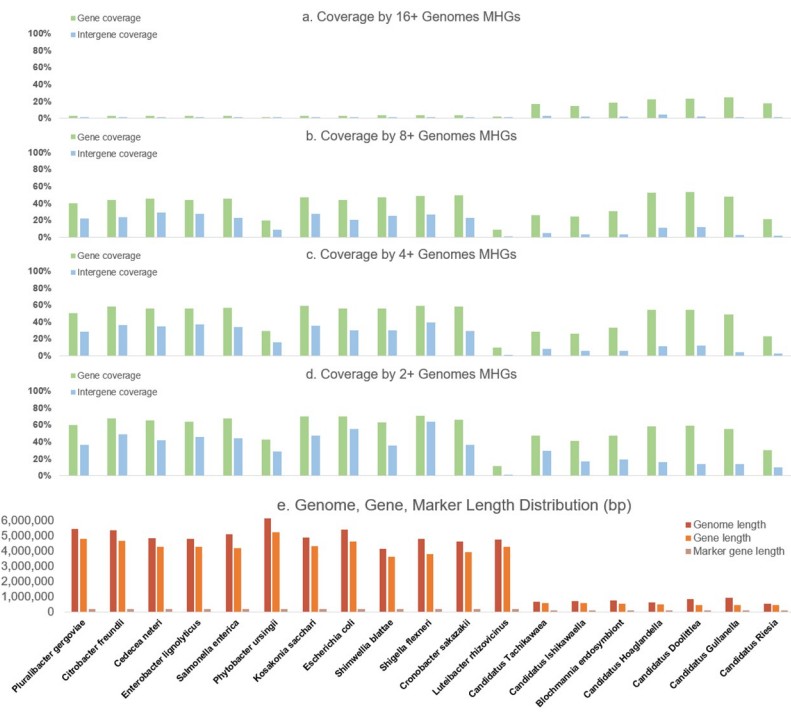

**Fig 4. Proportion of genes and intergenic regions covered by MHGs and genomic coverage by marker genes.** In subfigures a, b, c and d, gene coverage refers to the percentage of sites annotated as belonging to a UTR or coding sequence that is covered by a MHG, and intergene coverage refers to the percentage of all other sites covered by a MHG.

with 2 or more genomes represented. This is due to the fact that *L. rhizovicinus* is highly divergent from the other genomes in our analysis, being from an entirely different family (Rhodanobacteraceae) within $\gamma$-proteobacteria. Nonetheless, the substantial coverage of genic sequences demonstrates the suitability of using MHGs for interfamilial analysis.

### *Escherichia coli* gene functionality analysis

In order to explore whether marker genes or genes which overlap MHGs are biased towards specific functions, we analyzed the functional categorization of both those kinds of genes in *Escherichia coli*. We observed substantial over and under-representation among many gene categories by the set of marker genes (Fig 5). The three categories of gene function most over-represented among marker genes were catalytic activity, small molecule binding, and carbohydrate derivative binding. The three categories most under-represented were plasma membrane, establishment of localization, and transporter activity. While genes overlapping highly conserved MHGs (those with sequences from 16 or more species) saw similar biases, the representation of all categories was mostly unbiased among fuller subsets of MHGs (Fig 5).

### Conflict between markers and MHGs

MHGs inferred using our algorithm should be homologous across the entire length of every sequence in the MHG, indels excepted. In contrast, marker genes may have undergone rearrangements. Our method inferred multiple MHGs for each marker gene (S1 Appendix) and since our simulation study showed that conflicting phylogenetic histories within a gene can

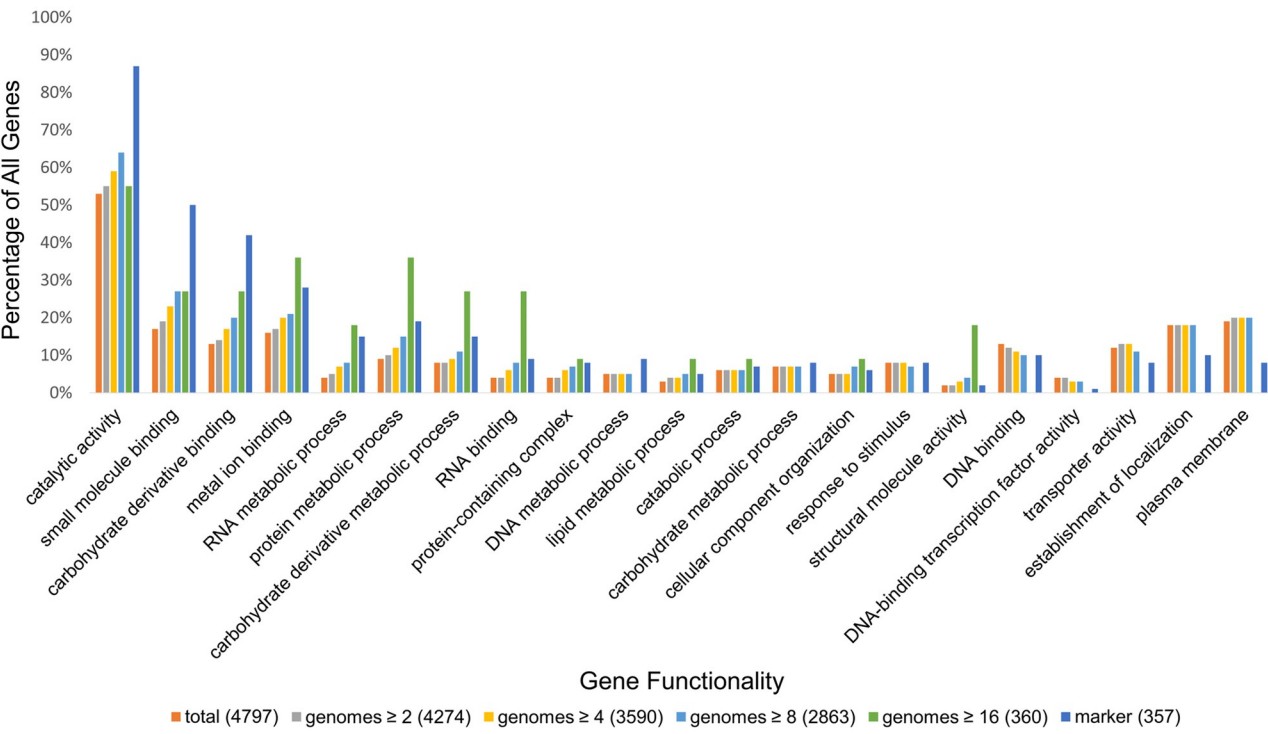

**Fig 5. *Escherichia coli* gene functionality coverage by marker genes and MHGs with $\geq x$ number of unique genomes.** We aggregated GO terms by AGR slim annotation, and only show percentages for terms that map to $\geq 1\%$ of *E. coli* genes. The total number of *E. coli* genes, and number of genes covered by markers, and the numbers of genes covered by MHGs with $\geq x$ number of unique genomes is given in brackets in the figure legend.

produce misleadingly high bootstrap support values, we compared the support values of splits from each MHG overlapping a marker gene with the support value of that split from the corresponding marker gene.

MHGs with fewer than four sequences were excluded because they contain no splits other than the tip branches, which are uninformative. Additionally, MHGs with more than one sequence for a single genome were excluded as this would make the analysis of support for phylogenetic splits intractable. Only 20.6% of MHGs with four or more sequences were excluded on that basis, and the remaining 79.4% of MHGs we term "single-locus informative MHGs."

For each marker-overlapping MHG, we considered the full set of splits observed in the entire bootstrap distribution for that MHG and the corresponding marker gene. Since the number of possible splits grows exponentially with the number of taxa, but only a few will be well-supported by the actual sequence alignments, a very long tail of splits with low MHG and/or marker gene support results (Fig 6). More interesting is that strong split support by marker gene alignments is not a good predictor of strong support by MHGs. When considering the marker genes with greater than 90% support, there appears to be almost no correlation with MHG support across the entire range of MHG support values (Fig 6). This is consistent with the aforementioned simulation study findings, further supporting the notion that high bootstrap support values derived from marker gene alignments may be misleading.

We also quantified phylogenetic (in)congruence using the extended quadripartition internode certainty (EQP-IC) metric [48]. Comparing to other statistics, EQP-IC aims at measuring the correctness of a given branch. It is robust when there are errors in the input reference tree.

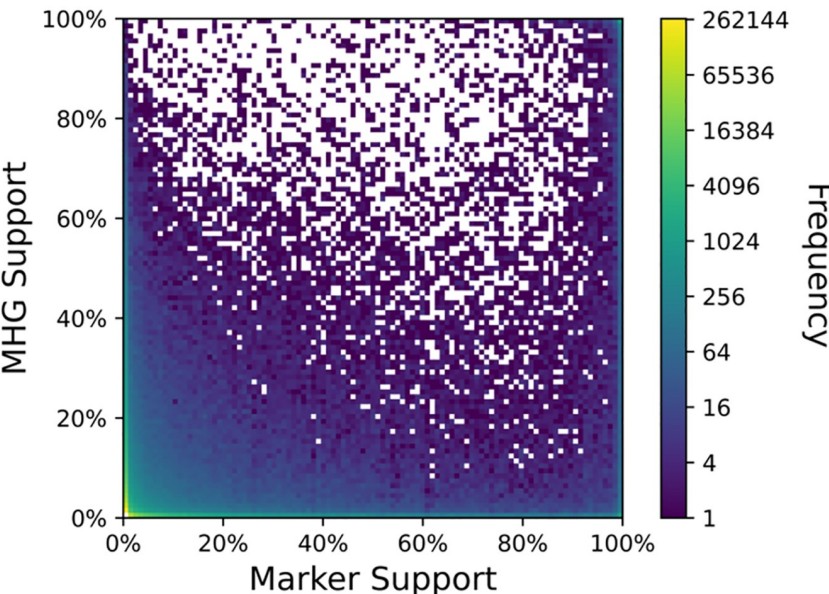

**Fig 6. Conflicting support for splits between markers and MHGs.** Bootstrapping was used to compute support. Each unique split from the MHG and marker gene tree bootstrap distributions, from each pair of MHG and corresponding marker gene, was counted once towards the frequency.

And in our case, EQP-IC values were calculated for each marker gene tree split, based on support by overlapping MHG gene trees. The modal value was around zero, indicating either a lack of information or a balanced mix of phylogenetic congruence and incongruence (Fig 7). However, given our observations of the number of highly supported splits, a lack of information is more likely (Fig 6).

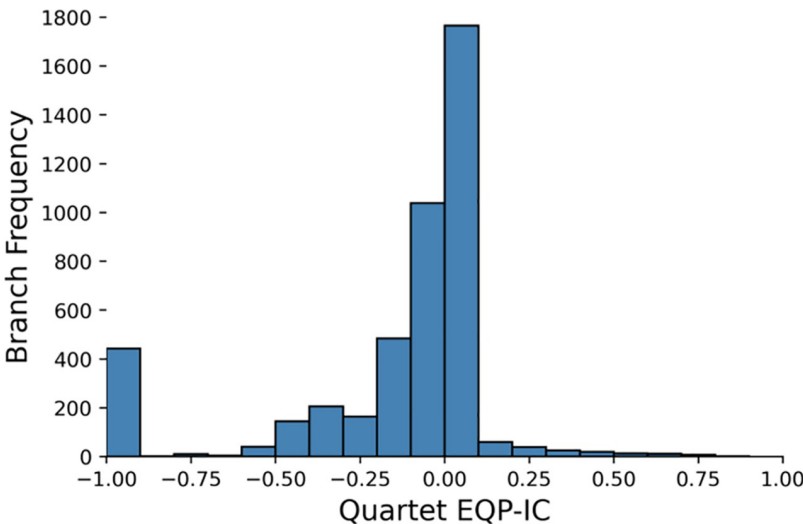

**Fig 7. Extended quadripartition internode certainty (EQP-IC) of MHGs overlapping marker genes.** Values closer to 1 indicates concordant phylogenetic histories between markers and MHGs, values closer to -1 indicate discordant histories, and values around 0 indicate either a lack of information, or a mix of concordance and discordance.

While very few marker splits were well supported by their overlapping MHGs according to this metric, a substantial number of splits were highly contradicted by those MHGs, with a spike in EQP-IC values observed around -1 (Fig 7). Values close to -1 indicate strong phylogenetic incongruence.

## Species tree comparison

There are many approaches to problems in comparative genomics, but a central concept that many other tools rely on, and which is used to explain and communicate the relationship among a set of species, is the species phylogeny. To understand whether the lack of coverage of genomes by marker genes, observable bias in functionality, and phylogenetic conflict with associated MHGs would impact the species phylogeny estimated from marker genes, we compared a species tree inferred from marker gene trees with a species tree inferred from MHG gene trees. For this analysis, we used all markers to build the marker species tree, but only used MHGs ≥ 400bp to build the MHG species tree. This was to exclude the effects of random error in shorter MHGs, as we found the distribution of bootstrap support for the branches of MHG gene trees ≥ 400bp was similar to markers, but the bootstrap support for the branches of more inclusive sets of MHG gene trees was generally lower (Fig 8). Again only MHGs with at least four taxa were included.

The method we used to infer species trees (ASTRAL-Pro) is able to use gene trees with multiple gene copies as input. Unlike markers genes, MHGs are not restricted to single copy sequences, so to understand if this changes the species tree result we ran ASTRAL-Pro with all MHGs ≥ 400bp and also with only single-locus informative MHGs ≥ 400bp, but the same topology was returned for both sets of MHGs (Fig 9).

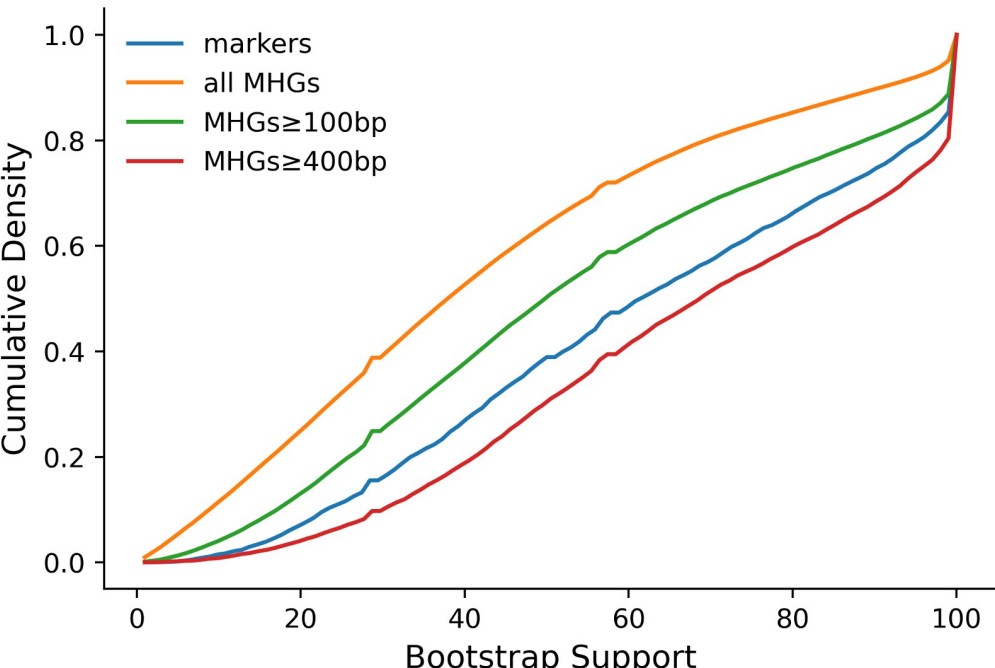

**Fig 8. Bootstrap support for splits.** The culmulative density distribution of bootstrap support is shown across all non-trivial splits from the maximum likelihood trees inferred from the original marker gene sequence alignments (blue). The same distribution is shown for the splits inferred from the original MHG sequence alignments (orange). In addition, the distribution of split bootstrap support from two restricted sets of MHG sequence alignments are shown; those with a length of at least 100bp (green) and those with a length of at least 400bp (red).

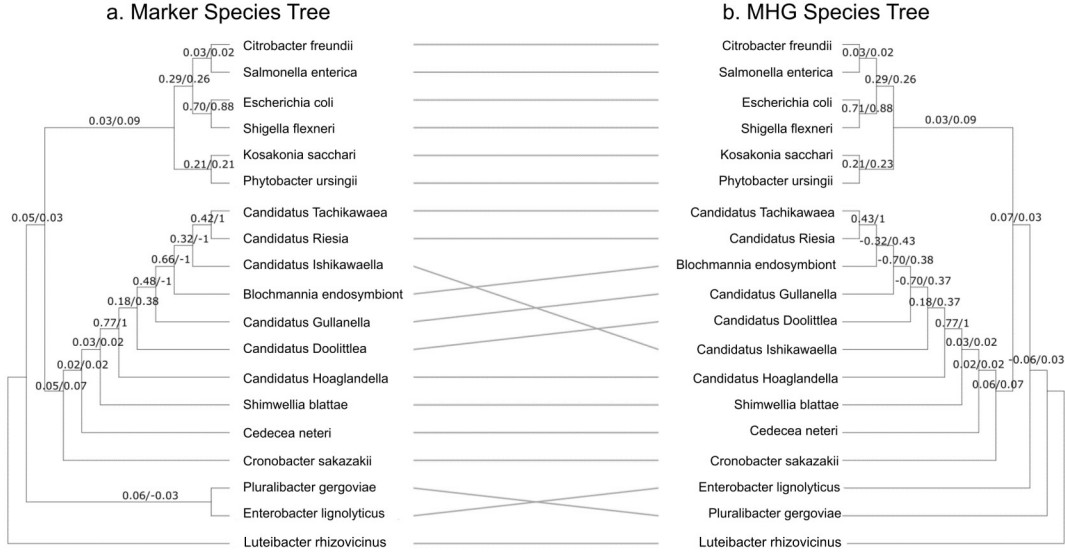

**Fig 9. Tanglegram showing the difference between the species tree inferred from MHGs ≥400bp in length compared with the species tree inferred from marker genes.** (a) Marker-based species tree. (b) MHG-based species tree. Support values are the extended quadripartition internode certainty (EQP-IC) from marker gene trees followed by the EQP-IC from ≥ 400bp MHG gene trees. Values closer to 1 indicates concordant phylogenetic histories between the gene and species trees, values closer to -1 indicate discordant histories, and values around 0 indicate either a lack of information, or a mix of concordance and discordance.

However, the MHG-based species tree topology differed in two ways from the marker-based topology. The more substantial difference is the placement of the endosymbiont *Candidatus* Ishikawaella, which the marker-based analysis groups with *Candidatus* Riesia and Tachikawaea. The EQP-IC support from marker genes for this grouping is strongly positive (0.66) indicating most marker genes containing relevant taxa provide support for this clade, but the EQP-IC support from MHGs is the most negative possible value (-1), indicating that virtually all MHGs containing relevant taxa reject it.

In addition, *Pluralibacter gergoviae* and *Enterobacter lignolyticus* are a clade in the marker-based species tree but not in the MHG-based species tree. The difference in EQP-IC support is however minor, with 0.06 support among markers and -0.03 support around MHGs, indicating a lack of signal or consensus for this grouping among both methods (Fig 9).

## Intergenic MHG analysis

Beside the potential bias introduced by an unrepresentative distribution of gene function, it is known that gene and intergenic sequences are subject to different evolutionary processes. For example, the rate of point mutations leading to non-synonymous amino acid changes may be substantially higher or lower than the rate for synonymous changes, depending on whether loci are under predominantly positive or negative selection [49]. This does not apply to intergenic sequences which definitionally do not encode for proteins even if they are functionally important. Marker genes do not include intergenic sequences, whereas our method does not exclude them, so we studied the possible effects of their inclusion.

We inferred species trees with EQP-IC support values from two subsets of MHGs. The first subset was restricted to MHGs which only contained sequences from inside genes, and the second subset was restricted to MHGs which only contained intergenic sequences. Because

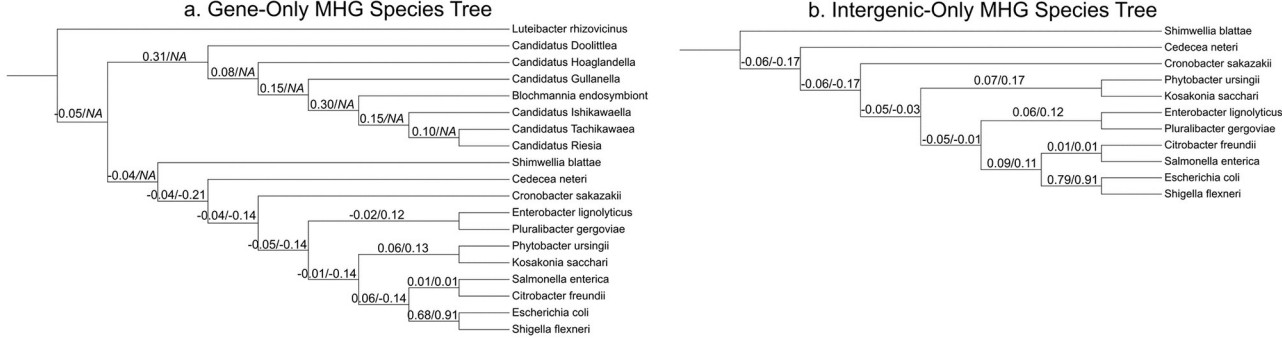

**Fig 10. Species trees inferred from gene-only and intergenic-only subset of MHGs.** (a) Species tree inferred from only MHGs entirely inside genes. (b) Species tree inferred from only MHGs entirely outside genes, which should be treated as unrooted since no outgroup taxa are present. Support values are the extended quadripartition internode certainty (EQP-IC) from marker gene trees followed by the EQP-IC from ≥ 100bp MHG gene trees. Values closer to 1 indicates concordant phylogenetic histories between the gene and species trees, values closer to -1 indicate discordant histories, and values around 0 indicate either a lack of information, or a mix of concordance and discordance.

homology is more difficult to identify between genomes at intergenic loci we reduced the MHG length threshold for this analysis from 400bp to 100bp. Still, the outgroup taxon *L. rhizovicinus* and all but one of the endosymbionts had no representation within intergenic-only MHGs. Furthermore, the remaining endosymbiont *Candidatus* Doolittlea was only included in two intergenic-only MHGs. For this reason, we excluded the outgroup and endosymbionts when estimating the intergenic-only MHG species tree and EQP-IC values.

After reducing the length threshold and excluding the two intergenic-only MHGs with *Candidatus* Doolittlea sequences, there were 3485 MHGs included in the first subset and 172 in the second, a similar ratio to the overall ratio of gene to intergenic genome content. The difference in topology between the gene and intergenic trees was an exchange of the *Phytobacter* and *Kosakonia* clade with the *Enterobacter* and *Pluralibacter* clade (Fig 10). This a "nearest-neighbor" interchange and as such is a minor difference. The support values for either arrangement is close to zero, indicating this change is likely sampling noise than a genuine difference between gene and intergenic evolutionary histories (Fig 10).

Reducing the length threshold appears to have reduced EQP-IC support, presumably because shorter MHGs have a lower information content and there is greater conflict between their splits. For either ≥100bp tree, the only non-endosymbiont clade supported by EQP-IC values was the *Escherichia* and *Shigella* sister relationship (Fig 10). Furthermore, empirically sound rooting of the intergenic-only tree was impossible since the outgroup was excluded, so we choose the root that best matches the gene-only MHG species tree, and this rooting should not be considered reliable.

## Discussion

The rapid advance in throughput and availability of sequencing technology has led to a deluge of genomic data. For bacteria, more than 430,000 assembled genomes were available as of April 2020 [50]. Studying the history of bacterial genome evolution offers an understanding of both the history of individual genes and operons, as well as broader evolutionary processes [51–54]. A medically significant implication is the acquisition of antibiotic resistance, both *de novo* and from other species [55, 56].

The complexity of genome evolution in bacteria is due to rampant horizontal gene transfer [57–60], gene duplication and loss [61], and *de novo* sequence evolution. Existing methods of

whole genome alignment work well at shallow timescales, but methods for studying evolution at deeper timescales in bacteria are hindered by this complexity, and rely on the assumption of non-recombination within marker genes. We have shown that this assumption, when broken, is not detectable or mitigated by low support values for branches affected by recombination. Not only that, marker gene methods rely on the accuracy of existing genome annotations. For newly sequenced genomes, annotations may not even be available [62].

By applying our method to 19 bacteria from a diversified sampling of Enterobacteriaceae, we have demonstrated its practical utility and more advantages over marker genes. Our method uses more sites from each genome, and is less biased in the functions of the corresponding genes that are included. Given a genome that is highly divergent from the rest, a family-level difference can be accurately captured by MHG. Instead of being limited by the constraint of high conservation as a marker gene, an MHG-based analysis is able to properly classify homologous relationships from any subset of the input genomes. We have shown that many highly supported splits from inferred marker trees are not supported by corresponding MHGs, concordant with our expectations from our simulation study where components have different phylogenetic histories. The EQP-IC measure provides evidence for where intragenic conflict affects the support for various taxonomic hypotheses, and for which clades are robustly inferred regardless of this conflict.

Of particular interest is our findings on the evolution of Enterobacteriaceae endosymbionts. Both marker and MHG phylogenies strongly support the monophyly of this group, in contrast to a previous concatenation analysis which claimed it has multiple origins [63]. These vertically transmitted endosymbionts provide essential amino acids to their insect hosts, and transmission may occur externally after oviposition [64, 65]. Neither marker nor MHG phylogeny is compatible with pure vertical transmission however, with the mealybug endosymbionts *Candidatus* Hoaglandella, Doolittlea and Gullanella being paraphyletic or polyphyletic. Based on the placement of those taxa, we hypothesize that this symbiotic relationship originated in mealybugs before diversifying to other insects. The major difference between phylogenies is that the MHG phylogeny is incompatible with this diversification occurring within a single lineage, as the EQP-IC measure strongly rejects a clade containing the four non-mealybug endosymbionts.

While the alignment graph enables some parallelization and space efficiency, improved scalability will be an important future direction for this algorithm and our implementation. The bottleneck of the current MHG tool is the polynomial (under ideal conditions) growth in runtime with the number of genomes. One avenue for alleviating this is replacing the all-vs-all BLAST with a subset of pairwise (genome by genome) BLAST queries, since MHGs are transitive in nature, complete connectivity between genomes is not required. This could use a phylogenetic guide tree, although rigorous study is needed to determine the best approach of choosing the subset of pairwise queries to minimize any loss of sensitivity. As with many algorithms, a divide-and-conquer strategy may also improve walltime performance, although not overall CPU time or power consumption. Beyond scalability, due to the stochasticity and heuristic nature of BLASTn, using the boundaries of BLASTn queries to define MHG boundaries may cause inferred MHGs to be shorter than necessary. This may be addressed in the future through post-processing, or by reconciling BLASTn query boundaries before or simultaneously with the partition and merge step.

Our work here demonstrates the fundamental problems of existing methods for prokaryotic phylogenetics at deeper timescales. Not only is it immediately useful on smaller, but still whole-genome, datasets, it provides the starting point for improved and scalable phylogenomic algorithms and implementations that are coherent with the actual complexity of the evolutionary histories across bacterial genomes.

## Supporting information

**S1 Appendix. Maximal homology group overlap of marker genes.** Each marker gene is labeled with the UniProt accession number of its reference protein. MHG sequences are drawn as colored arrows overlapping the sequences from each species for each marker gene. MHG sequences drawn with the same color belong to the same MHG, although colors are repeated when more than 20 MHGs overlap a marker gene. Arrowheads indicate the relative orientation of MHG sequences to show inversions.
(PDF)

## Acknowledgments

The authors would like to thank Todd J. Treangen and Vicky Yao for valuable feedback, and Bryce Kille for insights on whole genome alignment.

## Author Contributions

**Conceptualization:** Yongze Yin, Huw A. Ogilvie, Luay Nakhleh.

**Data curation:** Yongze Yin.

**Formal analysis:** Yongze Yin, Huw A. Ogilvie.

**Funding acquisition:** Luay Nakhleh.

**Investigation:** Yongze Yin.

**Methodology:** Yongze Yin, Huw A. Ogilvie, Luay Nakhleh.

**Project administration:** Luay Nakhleh.

**Resources:** Luay Nakhleh.

**Software:** Yongze Yin.

**Supervision:** Huw A. Ogilvie, Luay Nakhleh.

**Validation:** Yongze Yin.

**Visualization:** Yongze Yin, Huw A. Ogilvie.

**Writing – original draft:** Yongze Yin, Huw A. Ogilvie, Luay Nakhleh.

**Writing – review & editing:** Yongze Yin, Huw A. Ogilvie, Luay Nakhleh.

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
