## [Decision Letter · Decision Letter 0]

10 Feb 2022

Dear Dr Ogilvie,

Thank you very much for submitting your manuscript "Annotation-free Delineation of Prokaryotic Homology Groups" for consideration at PLOS Computational Biology.

As with all papers reviewed by the journal, your manuscript was reviewed by members of the editorial board and by several independent reviewers. In light of the reviews (below this email), we would like to invite the resubmission of a significantly-revised version that takes into account the reviewers' comments.

We cannot make any decision about publication until we have seen the revised manuscript and your response to the reviewers' comments. Your revised manuscript is also likely to be sent to reviewers for further evaluation.

Sincerely,

Rachel Kolodny

Associate Editor

PLOS Computational Biology

Jian Ma

Deputy Editor

PLOS Computational Biology

Reviewer's Responses to Questions

**Comments to the Authors:**

Reviewer #1: Yin et al. introduced an algorithm for identifying maximal homologous groups of sequences (MHGs) from prokaryotic genomes. The method starts with pairwise alignments among genomes, and piles up homologous segments using a network approach to generate longer MHGs. They suggested that MHGs are advantageous for phylogenetic inference over traditional marker genes which are prone to the impact of horizontal gene transfer and recombination events. The MHG method they developed does not rely on the annotation of protein-coding genes. They were shown to cover higher proportion of the genomes and be less biased in terms of function. The authors applied this new method to 19 Enterobacteriaceae species previously analyzed in a marker gene-based phylogenomic study. They found that MHGs granted higher accuracy and robustness compared with markers. Collectively, they recommended MHGs for phylogenomic studies of prokaryotes.

This is a very interesting method and direction of phylogenetic research. It provides new evidence for a plausible solution to the philosophic question of the unit of evolution. It comes with a software tool, in addition to an algorithm, which can be used by the research community. This work is potentially impactful to the field of phylogenetics. The authors concluded that it provides a “starting point” for future researches along the line, which I largely agree and appreciate. I will be amazed to see how gene-free phylogenetics will expand in the future, and I believe that the current work sets an important milestone for it.

With that being said, I think the current work is still quite primitive in the breath and depth of the analysis. Admittedly, phylogenetics is a large question, and it shouldn’t be a requirement for researchers with novel thoughts to take care of all plausible questions before publishing their result. I still would like to point out some of the questions which I think are most relevant, important and “good-to-have”, and hope that the authors can address these without substantial amount of additional efforts.

One potential mechanismal problem of the use of MHGs, as I would propose, is that MHGs can span over multiple genes and intergenic regions of the genome, and these nucleotide elements may not follow the same evolutionary model. Current phylogenetic methods, such as maximum likelihood (as the authors used), assume that all sites in an MSA can be modeled using a distribution of mutation rates. If there are obvious heterougenous regions in the MSA, the typical method is to “partition” the MSA and apply separate evolutionary models to individual partitions. In the MHG study, this was apparently not considered, and consideration of partitions will revert MHG back to traditional marker-based protocols. I will be curious to see a comparative analysis of MHGs splitting into gene / intergenic regions and how that impacts phylogenetic inference.

One potential disadvantage of this method, as I would imagine, is that it may struggle with a set of taxa with high divergence, because nucleotide-level homology is hard to identify using BLASTn when the evolutionary distance is large. In this work, the authors only tested the method of a family-level taxon set. They mentioned in line 77: “when the divergence is low enough for sequence homology to be identifiable at a nucleotide level…” I will be curious about the performance in more diverse datasets, such as order- or class-level taxon sets. It may or may not be necessary to add new analyses, but some discussion should be favorable.

One thing the authors haven’t discussed (only briefly mentioned in the Discussion (line 380) is the scalability of this algorithm. Because it relies on all-vs-all BLAST, I guess that the computational complexity is at least O(n^2). I futher guess that alternatives like Usearch and Minimap can accelerate this process, although may or may not change the scaling factor. The test dataset only has 19 genomes, which is a very small one in modern phylogenomics. The readers may be curious about how many taxa can they include in such an analysis. Even further optimization is not introduced in the current work, I would suggest that the authors at least make some discussion and point to future directions.

In the simulation study, the accuracy of the phylogenetic trees was evaluated using the following (line 259): “For the non-recombining segments, bootstrap support for true branches was higher than incorrect branches, suggesting that bootstrap support values for inferred MHG trees can be used to identify true splits.” While I can understand its logic, I do not see this as a compelling approach. Higher bootstrap from non-recombining segments is largely anticipated, but whether this can reflect “true” evolutionary history is a totally independent question. Use of semi-real datasets (as in contrast to purely simulated dataset) for validating a phylogenetic approach does have the problem of not being able to identify the grouth truth, and I doubt if the current solution is a complete solution. One thing that would be good to add is to introduce some biology – Enterobacteriaceae is a well-studied taxonomic group, with numerous literatures discussing its classification and evolution. The authors may read some literatures and compare the phylogenetic trees (marker- or MHG-based) with the knowledge base. If they can identify some well-known evolutionary pattern supported by one tree but not the other, it could be interesting and informative to the readers.

With regard to the terminology – is “homologous” in MHG actually “orthologous”? Because HGTs and recombinations within the same gene family (i.e., same evolutionary origin, despite old or young) do not break homology; they are just not orthologous which is the assumption for tree building.

Line 35: There are two “even”s.

Finally, I suggest that the authors make more description of the software tool they developed. I checked the GitHub repository and found that it has a decent command-line interface and documentation. seemingly ready for end users. If that’s the case, the audience may want to know because they will be prone to give it a try.

Reviewer #2: In the manuscript titled "Annotation-free delineation of prokaryotic homology groups", Yin et al. present a pipeline for annotating sequences to identify maximal homologous group (termed MHGs). These MHGs can be interpreted as non-recombining blocks (or at least no evidence of recombination from the sequences), which is important for estimating gene trees with high fidelity and which has downstream impacts on understanding species relationships at the genome scale. Appropriate non-recombining blocks are particularly difficult to identify for genomes separated by deep timescales, as may be found through comparison of highly divergent prokaryotic taxa.

The manuscript is easy to understand, with clear visualizations describing the authors' proposed pipeline, which makes it accessible to the non-expert computational biologist. The authors also released open-source software on GitHub, ensuring that their method is easily applied by the broader scientific community. Overall, I believe this is a quality study. I have mostly minor concerns that I list below, which relate to certain analysis, presentation, or experimental choices

1. On lines 199-201, the authors should motivate their choices for the BLASTn parameters either by citation or by providing their reasoning.

2. On lines 208-210, for simulation of marker genes, the component gene trees were obtained by performing a nearest-neighbor interchange (NNI) operation on the immediately preceding component tree (or initial tree if first component gene tree). My understanding is that subtree prune and regraft (SPR) is a better model of recombination than is NNI. Why not use SPR here instead of NNI? If SPR would make identification of MHGs more difficult, then I think it would be worth understanding performance under such a setting.

3. On lines 211-214, the authors simulated 15 component gene trees that were then used to evolve 100 base sequence alignments to generate an overall sequence of length 1.5 kb. These sequences evolved with equal transition rates and base frequencies. What is the motivation for these particular parameter choices? Are these consistent with the empirical results obtained from the prokaryotic data? That is, are the transition rates and base frequencies close to genome expectations from the empirical data, and are 1.5 kb sequences a reasonable size to understand method performance from the simulated data? Motivation for these parameters or citations should be provided to ensure that the simulated setting is realistic.

4. On line 232, internal branch lengths less than 0.0005 were treated as unresolved. What is the motivation for this number? Is it based on fewer than one mutation across the 1.5 kb sequence? That is, 1/1500 is approximately 0.00067, which is maybe how 0.0005 was derived?

5. On lines 276-280, coverage information is highlighted relating to K. michiganensis being poor. However, from Figure 4, it appears as if roughly half the sampled species in the empirical analysis have similarly poor coverage compared to K. michiganensis. Because of this, I am unsure why K. michiganensis is specifically being highlighted here.

6. In Figure 4, why is gene length almost the same size as genome length? Is this because the intergenic regions are small, and so the genome is fairly gene dense?

7. In Figure 5, the y-axis should read "Percentage of all genes" rather than proportion. Also, how are the GO categories on the x-axis ordered? Why not order them by over- to under-representation of markers, so it is easier to follow the way at which the results are described in the text for this figure.

8. In Figure 8, why not plot cumulative distribution functions (CDFs) instead of estimated density plots derived from smoothing kernels? This approach would encode all the necessary information to compare the distributions, without having to smooth the data with a kernel density estimator. That is, we will see the actual data rather than smoothed data.

Reviewer #3: This paper is a scholarly attempt to address the problem of phylogenetic inference in the case where different genes and gene fragments have different evolutionary histories, due to a variety of mechanisms by which DNA can be transmitted non vertically. It shows in particular that marker gene methods are quite biased and can easily end up including genes with conflicted signals that can bias the results. The paper is honest about its limitations, which I take to be a lack of scaleability.

My first main comment is that the paper shows how the method works in principle to identify genome alignments, but does little to show us what this looks like in practice. I.e. what do MHG blocks look like in practice for the real data as opposed to the conceptual data shown in figure 1. This may be difficult to look at for the full 19 species dataset, in which case a smaller one could be used for this purpose.

My second main comment is that the section "correspondence between markers and MHGs" and the two main text figures associated with it (figures 6 and 7) is hard to understand. Full concordance of genes seems perhaps unrealistic in a dataset with 19 taxa, while large parts of the tree are likely to be concordant between every methods. So it is easy to quantify concordance in a way that is either over-sensitive to small differences or is insensitive to fairly substantial ones. I find this section very technical and do not have too much orientation about what individual datapoints mean. Maybe there is a way of choosing a simplified example, either with less taxa, and explaining the metrics in more concrete terms to make it clearer what is going on.

**Have the authors made all data and (if applicable) computational code underlying the findings in their manuscript fully available?**

Reviewer #1: Yes

Reviewer #2: Yes

Reviewer #3: Yes

PLOS authors have the option to publish the peer review history of their article (what does this mean?). If published, this will include your full peer review and any attached files.

Reviewer #1: No

Reviewer #2: No

Reviewer #3: No
---

## [Decision Letter · Decision Letter 1]

16 May 2022

Dear Dr Ogilvie,

We are pleased to inform you that your manuscript 'Annotation-free Delineation of Prokaryotic Homology Groups' has been provisionally accepted for publication in PLOS Computational Biology.

Best regards,

Rachel Kolodny

Associate Editor

PLOS Computational Biology

Jian Ma

Deputy Editor

PLOS Computational Biology

Reviewer's Responses to Questions

**Comments to the Authors:**

Reviewer #1: Yin et al. made high-quality responses to the comments raised by me and other reviewers, as well as rigorous revision of the manuscript. They added a new analysis to compare genic vs. intergenic MHGs in phylogenetic inference. This analysis somehow addressed my question about gene partitioning -- not precisely, but at least relieved some related concerns. The added discussion on L. rhizovicinus does address my question on distantly related taxa. However, it is merely a single case instead of a systematic exploration. But I agree with the authors, and stay curious about their future works in this direction. I appreciate the added paragraphs discussing the time complexity of the algorithm, which is useful to the audience. The guide tree and divide-and-conquer solutions proposed by the authors sound viable to me. I will be intrigued to learn about their future improvement. My own guess is that MinHash-based approaches can be a first pass to exclude genome pairs that are less likely to share MHGs. I also appreciate the addition of the Enterobacteriaceae endosymbionts discussion. Although this does not distinguish the MHG tree from the marker tree, but the discussion of a biological question showed that the authors did put thoughts into their findings. Overall, I think the authors have made valuable efforts in resolving my concerns. Although some questions are big and worth future independent works, at this moment, I don't see reason to block the publication of this high-quality work, which will likely inspire the phylogenetics community.

Reviewer #2: Overall I think this is an improved version of an already nice manuscript.

I only have one remaining comment, which is that the y-axis on Figure 5 still reads as "Proportion of All Genes" when it should read as "Percentage of All Genes" since the tick marks are labeled as percentages.

**Have the authors made all data and (if applicable) computational code underlying the findings in their manuscript fully available?**

Reviewer #1: Yes

Reviewer #2: None

PLOS authors have the option to publish the peer review history of their article (what does this mean?). If published, this will include your full peer review and any attached files.

Reviewer #1: No

Reviewer #2: No

---

## [Editor Report · Acceptance letter]

3 Jun 2022

PCOMPBIOL-D-21-02243R1 

Annotation-free Delineation of Prokaryotic Homology Groups

Dear Dr Ogilvie,

I am pleased to inform you that your manuscript has been formally accepted for publication in PLOS Computational Biology. Your manuscript is now with our production department and you will be notified of the publication date in due course.

With kind regards,

Zsofia Freund
